# Qualitative assessment of hand hygiene knowledge, attitudes and practices among healthcare workers prior to the implementation of the WHO Hand Hygiene Improvement Strategy at Faranah Regional Hospital, Guinea

**Moussa Douno**[1]*, **Carlos Rocha**[2], **Matthias Borchert**[2], **Ibrahima Nabe**[3], **Sophie Alice Müller**[2]

**1** Projet des Fièvres Hémorragiques en Guinée, Centre de Recherche en Virologie, Université Gamal Abdel Nasser de Conakry, Conakry, Guinée, **2** Center for International Health Protection, Robert Koch Institute, Berlin, Germany, **3** Hôpital Régional de Faranah, Faranah, Guinée

\* msokadouno@gmail.com

**Data Availability Statement:** We are not able to make our underlying data set publicly available for

## Abstract

Healthcare-associated infections are a serious burden globally. Few qualitative studies have explored healthcare workers' knowledge, attitudes and practices of hand hygiene. Prior to the implementation of the World Health Organization's Hand Hygiene Improvement Strategy at Faranah Regional Hospital in the Upper Region of Guinea in December 2018, we conducted a qualitative baseline assessment of knowledge, attitudes and practices of hand hygiene among healthcare workers to guide future hand hygiene interventions. The qualitative study consisted of direct observations, In-Depth Interviews (IDIs) and Focus Group Discussions (FGDs). We found that the 2013–16 Ebola outbreak had had a pivotal impact on healthcare workers' knowledge, attitudes and practices. The severity of the disease and the training provided for infection control were responsible for their knowledge acquisition and adoption of good attitudes and practices. However, negligence, resulting in poor hand hygiene practices, rose after the outbreak, once the "cue of fear" that had motivated workers for their own self-protection had waned. Our results suggest that local capacity building through training and availability of hand hygiene materials would be a sustainable approach to enhance hand hygiene culture at the hospital. Our study suggests that there is a need for a high and long-term commitment of authorities and healthcare workers at all levels for a sustainable hand hygiene culture.

## 1. Introduction

Hand hygiene best practices refer to the task of cleaning hands at "the right times and in the right way" [1], and this task, when optimally performed, plays a pivotal role in the prevention

ethical and data protection reasons. The data contain potentially identifying information: our data have been collected from a small group of participants, and even data that are not directly identifying in combination become identifying (e.g. sex, profession, department ward, work description). This restriction to data availability has been imposed by RKI's Data Protection Office. Data requests may be sent to the ZIG Department Secretariat via ZIG-Assistenz@rki.de.

**Funding:** This study was funded by the BMZ (Bundesministerium für Zusammenarbeit) as part of the GIZ (Gesellschaft für Internationale Zusammenarbeit) University and Hospital Partnerships in Africa (ESTHER) Program (Ensemble pour une Solidarité Thérapeutique Hospitalière en Réseau) (Award Number 81213469). MB received this grant. The funders had no role in study design, data collection and analysis, decision to publish or preparation of the manuscript.

**Competing interests:** The authors have declared that no competing interests exist.

of healthcare-associated infections (HCAIs) and contributes to the improvement of patients' outcomes [2]. According to the World Health Organization (WHO), "the burden of healthcare-associated infections (HCAIs) worldwide is very high in terms of morbidity, mortality, extracosts, and other outcome indicators" [3]. In low- and middle-income countries, this burden is even higher due to the limited awareness of these infections and the prioritization of other health issues over patient safety, for example controlling endemic diseases, such as HIV, and meeting the health-related Millennium Development Goals (MDGs) [4, 5]. Healthcare-associated infections can be reduced by up to 55% if healthcare workers (HCWs) know and apply appropriate hand hygiene techniques [3, 4]. Seeking to lower the high incidence rate of healthcare-associated infections, the WHO launched its Multimodal Hand Hygiene Improvement Strategy in 2009 [1].

The WHO strategy was first implemented on the African continent in 2006 [1, 6–8], and may have been spurred in part by the 2013–16 Ebola Virus Disease (EVD) outbreak. Hand hygiene knowledge and compliance were particularly low in settings with limited resources and poor infrastructure in East and West Africa [6–8]. Certain professional categories such as doctors or nursing assistants, types and intensity of patient care, hospital wards, as well as understaffing and high workload, have been considered to be some of the factors that affect the compliance with hand hygiene [9]. At Faranah Regional Hospital (FRH), the implementation of the WHO strategy led not only to high levels of knowledge and compliance but to a reliable supply of alcohol-based hand rub [10]. A long-term follow-up mixed methods study at the hospital showed, however, that knowledge and compliance waned over time. The study found that healthcare workers took pride in the local production of alcohol-based hand rubs, but that they felt they needed further training on hand hygiene [11]. The latter need was expressed even though the majority of the hospital's healthcare workers had been trained during and after the Ebola outbreak [10]. Healthcare workers' request for further training could be motivated by the fact that the training sessions during the epidemic were associated with some advantages such as per diems that were given to trainees.

In recent years, qualitative research has increasingly been recognized as a relevant methodology in research of evidence-based practices and health services [12–14]. Its potential to provide good quality and in-depth data about hand hygiene guidelines and the implementation of protocols has increased the interest of researchers to apply these methods in research on infection prevention and control [15–19]. However, little qualitative research has been published about baseline assessments conducted prior to the implementation of the WHO Hand Hygiene Improvement Strategy in low- and middle-income countries and has instead focused on the exploration of factors favoring or hindering the use of alcohol-based hand rubs [20].

The main goal of this study was to generate qualitative insights into the knowledge, attitudes and practices of healthcare workers about hand hygiene in order to inform and guide the implementation of the WHO hand hygiene improvement strategy at FRH. This qualitative data offers a baseline for follow-up research and, critically, for the planning and implementation of valid and impactful interventions at FRH. The findings are relevant not only in a post-Ebola setting for a better preparedness for epidemic diseases, but also for the prevention of healthcare associated and postoperative infections in hospital settings [21, 22].

## 2. Methods

### 2.1. Study design and period

This study is embedded in the Partnership to Improve Patient Safety and Quality of Care (PASQUALE), a project that, in its first phase, tackles the first WHO Global Patient Safety Challenge: "Clean Care is Safer Care" [23], including the implementation of the WHO Hand

**Table 1. The project timeline.**

| Year | 2017 | | | 2018 | | | | | | | | | | | | 2019 | | | | | | | | | | | |
|---|---|---|---|---|---|---|---|---|---|---|---|---|---|---|---|---|---|---|---|---|---|---|---|---|---|---|---|
| Month | 10 | 11 | 12 | 1 | 2 | 3 | 4 | 5 | 6 | 7 | 8 | 9 | 10 | 11 | 12 | 1 | 2 | 3 | 4 | 5 | 6 | 7 | 8 | 9 | 10 | 11 | 12 |
| Baseline qualitative | | | X | | | | | | | | | | | | | | | | | | | | | | | | |
| Baseline quantitative* | | | | X | X | X | | | | | | | | | | | | | | | | | | | | | |
| Intervention* | | | | | | | | | | | | | | | X | | | | | | | | | | | | |
| Follow-up 1 quantitative* | | | | | | | | | | | | | | | X | X | X | | | | | | | | | | |
| Follow-up 2 quantitative and qualitative* | | | | | | | | | | | | | | | | | | | | | X | X | | | | | |

*These steps have been examined in separate publications [10, 11].

Hygiene Improvement Strategy and the local production of alcohol-based hand rubs at FRH. The project, funded by the GIZ ESTHER Alliance (*Ensemble pour une Solidarité Thérapeutique Hospitalière en Réseau*), followed a holistic approach, with quantitative and qualitative data collected in several rounds of research (Table 1). Whereas the quantitative results are reported elsewhere [10], here we report on the qualitative data of the pre-intervention baseline assessment. We conducted a one-month qualitative study in December 2017, comprising direct observation of healthcare workers in their daily activities at the hospital and semi-structured in-depth interviews (IDIs) and focus group discussions (FGDs) with them.

## 2.2. Study setting

FRH is a governmental reference hospital located in the town of Faranah, in Faranah prefecture of the administrative region of Central Guinea. It is a partner hospital of the Robert Koch Institute in Berlin, Germany. It serves a population of 300,000 inhabitants mostly of the Malinke ethnic group [24]. At the time of writing, the hospital employed 74 healthcare and administrative staff and comprised 13 wards, including surgery, laboratory and, in response to the 2013–16 Ebola outbreak, a center for the treatment of epidemic diseases.

## 2.3. Study population

The study population consisted of medical doctors, nurses, midwives, nursing assistants and laboratory assistants at FRH. The main inclusion criteria were being a healthcare worker in one of the hospital's wards and willing to participate in the study. All participants were purposively sampled among available staff to represent various hospital units, professions and both genders [25]. Thirteen in-depth interviews (IDIs) were conducted in the thirteen hospital wards with healthcare workers (one per ward–mainly nursing and laboratory staff), and twenty-five people participated in three focus group discussions (FGDs) (one with doctors, one with midwives and one with nurses/nursing assistants/laboratory technicians). In total, the study population comprised 38 hospital workers, of which two thirds were between 20 and 40 years of age and had more than 6 years of practice (66%), while the gender distribution was roughly equal (Table 2). Medical doctors, midwives and nurses all contributed at least a quarter of the study population each.

## 2.4. Data collection

Data collection was carried out at the hospital by one of the authors [MD] who is a medical doctor and familiar with the hand hygiene strategy. He was not known to the participants. Each participant was provided with a thorough explanation of the study objectives and then invited either to an individually arranged interview or to a group discussion, depending on their daily service duties. The interview guides comprised three sections which sought to

**Table 2. Study population and characteristics.**

| Characteristics | N (38) | % |
|---|---|---|
| Age range (years) | | |
| 20–29 | 10 | 26 |
| 30–39 | 15 | 39 |
| 40–49 | 7 | 18 |
| 50–59 | 4 | 11 |
| 60 and more | 2 | 5 |
| Gender | | |
| Male | 20 | 53 |
| Female | 18 | 47 |
| Profession | | |
| Medical Doctors | 10 | 26 |
| Midwives | 9 | 24 |
| Nurses/nursing assistants/laboratory technicians | 19 | 50 |
| Years of experience | | |
| 0–2 | 5 | 13 |
| 3–5 | 8 | 21 |
| 6–10 | 11 | 29 |
| more than 10 | 14 | 37 |

explore the healthcare workers' hand hygiene-related knowledge, attitudes and practices at the hospital. IDIs were mainly conducted with nursing and laboratory staff as they are more involved in patients' care and sample handling, some of the acts that request more hand washing. One FGD was separately performed with each professional group (doctors, midwives, and nurses/nursing assistants/laboratory technicians) in order to avoid hierarchical influence due to differences in levels and positions between doctors and their subordinate nurses/technicians, and then encourage discussions among peers with almost the same levels and positions. Those who were individually interviewed did not participate in FGDs. All IDIs and FGDs were audio recorded in the presence of only the facilitator and participants with an average of 45 minutes for both. Furthermore, direct observations were also performed. At the outset of the study, the hospital's director introduced MD to the staff as a peer who had come to conduct the assessment in preparation for the start of the PASQUALE project, inviting them to welcome him in their various wards to facilitate the study. This facilitated MD's integration in the wards, though it might have influenced his interlocutors' behavior during the data collection period. After familiarizing himself with the personnel, MD performed three main observations by attending to: 1) an inpatients' medical visit in the surgical ward led by a doctor with his team composed of nurses/nursing assistants and many interns–both medical and nursing students (more than a dozen); and 2) two different sessions in which two nursing assistants provided care to outpatients in pediatrics and general medicine wards. Patients were not informed for the observations as they were performed during medical routines to which MD attended as a peer. Observations were limited to whether or not hand hygiene was performed and did not include an assessment of the correctness of the procedure. Notes were taken on observed actions after leaving the observation sites.

## 2.5. Data analysis

Sociodemographic data of the respondents were descriptively summarized in means (see Table 2). By assigning a code to each of the thirteen IDIs and three FGDs, participants were

de-identified. Audio recordings were transcribed verbatim, and all transcripts were cross-checked for consistency to enable thematic analysis. The initial analytical coding frame was developed following the main points of the topic guide and was performed using Microsoft Word for Windows 10 (deductive coding). The recurrent themes that emerged from the data were used to refine the initial codebook (inductive coding). Based on that, a thematic content analysis was then conducted through line-by-line reading of all the transcripts, and the results were informed by concepts grounded in the data. Notes from the observations were descriptively elaborated to form a sub-section of the study results. Fieldwork, transcription and analysis were carried out in French; verbatim quotations were translated into English by MD for the purpose of this paper.

## 2.6. Ethics

Ethics approval was obtained from the *Comité National d'Ethique pour la Recherche en Santé*, Guinea. Permission to conduct the study was granted by the local health authorities, and every participant received information about the study and signed a written consent form. Verbal consent was obtained from the heads of units for direct observations; no consent was sought from patients as the researcher (himself a medical doctor) attended the observed activities as a peer.

## 3. Results

The results are presented in terms of three categories: first, the knowledge healthcare workers had of hand hygiene; second, the attitudes healthcare workers had toward hand hygiene; and third, hand hygiene practices. Differences in these three categories based on healthcare workers' professional backgrounds were not considered in this study.

## 3.1 Knowledge

**3.1.1 Preventing healthcare-associated infections.** The healthcare workers recognized hand hygiene as a fundamental and indispensable means to prevent the spread of infection within the hospital environment. They believed that hand hygiene was key to stop the spread of healthcare-associated infections, and therefore ensure the safety of both personnel and patients. Some emphasized the importance of hand hygiene in preventing infection even beyond the hospital setting:

> "Hand hygiene allows protecting providers, the community and humanity as a whole by avoiding nosocomial infections". (Doctor, FGD)

**3.1.2 Providing quality care to patients.** Healthcare workers mentioned the different methods they knew and used for hand hygiene in their health facilities, ranging from washing hands with water and soap, using a 0.05% chlorine solution, to employing alcohol-based hand rubs. They described the situations when hand hygiene should be performed: before and after each medical procedure, before wearing gloves and after removing them, after contact with a patient and/or their environment, before and after exposure to biological fluids, and after an invasive procedure (blood taking, tube placement, etc.). Some affirmed that compliance with these methods enables the provision of quality care to patients, thus establishing a climate of trust between providers and clients and increasing the credibility of the hospital. They also believed that this practice contributes to the reduction of morbidity and mortality in their health facility.

"The importance of hand hygiene is to prevent contagious diseases, to protect ourselves and our patients from these contagious diseases, [. . .] which allows us to offer them quality care". (Nurse, FGD)

**3.1.3 Learning through the Ebola outbreak.** Participants affirmed that it was in the context of the fight against the 2013–16 Ebola epidemic that they had learned more about hand hygiene through training received from various partners involved in the crisis response. They posited that it was the correct application of this acquired knowledge that made it possible to curb the epidemic in the country.

"Before [the Ebola epidemic], we used to consult patients without washing or disinfecting our hands. Ebola came to unveil these weaknesses, there were really deaths. Since we adopted these good practices, these phenomena are really starting to disappear". (Doctor, FGD)

In summary, healthcare workers showed broad knowledge of hand hygiene in their discussion of the role it plays in preventing healthcare-associated infections, stopping the spread of epidemics and reducing mortality. Hand hygiene training offered during the Ebola outbreak constituted a major source of acquired knowledge amongst them. The main methods they knew for efficient hand hygiene were water and soap and a 0.05% chlorine solution, which was the standard hand disinfectant during the epidemic, and they considered these to have contributed significantly to curbing the 2013–16 Ebola epidemic.

## 3.2 Attitudes

**3.2.1 The influence of the Ebola outbreak.** Perceiving hand hygiene as a means of protection against infections influenced healthcare workers' attitudes toward implementing hand hygiene. From FGDs, it appeared that positive attitudes toward hand hygiene were rooted in healthcare workers' experience of the 2013–16 Ebola outbreak that, according to the respondents, was spread because of poor attitudes toward hand hygiene within the hospital environment and in the wider community.

"This change of attitude is due to the Ebola outbreak. I think it was an evil that opened the minds of Guineans". (Doctor, FGD)

Among measures put in place during this crisis, the existence of hand hygiene protocols in the different wards of the hospital was reported to have had a great influence on healthcare workers' attitudes toward hand hygiene. According to the study participants, these protocols explained the steps of hand hygiene with photos or posters, forming a regular reminder to uphold hand hygiene when completing the daily duties. The posters on the walls also reminded them of the different steps and techniques involved in each of the hand hygiene methods.

"Thanks to these protocols and posters we always see on the walls, we have considerably changed our attitudes in terms of hand hygiene. . ..". (Doctor, FGD)

Participants explained that the Ebola epidemic had led to the institution of training for healthcare workers on hand hygiene and infection prevention and control, the elaboration of protocols on hand hygiene, mass sensitization on the need for hand hygiene, the provision of necessary materials for the implementation of hand hygiene practices, and supervision in the

health facilities. The respondents believed that all these elements contributed to reinforcing the development and maintenance of a good attitude toward hand hygiene within the hospital.

The Ebola outbreak played a major role in changing the attitudes of healthcare workers at FRH toward hand hygiene. This outbreak and its related preventive measures–mainly hand washing–turned this practice into an automatic reflex for healthcare workers who were concerned with saving not just their patients' but their own lives.

**3.2.2 Hand hygiene neglectful attitudes.** Despite the protective role of hand hygiene and changes in attitudes acquired during the 2013–16 Ebola outbreak, the participants spoke to attitudes of negligence toward hand hygiene. Some participants, particularly doctors, reported that a few months after the end of the Ebola epidemic in the country they began to note neglectful attitudes among healthcare workers who would ignore hand hygiene protocols at the hospital. While some believed that the negligence arose because healthcare workers considered the risk of being infected with Ebola was over and because there was a shortage of hygiene materials, others blamed it on healthcare workers who failed in sustaining positive attitudes towards hand hygiene acquired during the Ebola epidemic.

"It's negligence [. . .], otherwise it's not something that's very hard to get. Everyone should have a bucket with a tap for hand washing before eating, after toilets and after meals, and so on. It is because I didn't play my role as a health worker. [. . .] Every health worker should make a difference in his and other families. [. . .] We have to create this difference, we as agents. So it's not a lack of means, it's a negligence from our side". (Doctor, FGD)

Some Participants even mentioned that the spread of the Ebola virus in health facilities was due to these neglectful attitudes toward hand hygiene practices, resulting in the death of numerous healthcare workers.

"It is because of the negligence of hand hygiene that Ebola killed some healthcare workers elsewhere. Here at this hospital, if you've noticed that among all the staff, there was no Ebola cases, it is thanks to hand hygiene. . .". (Nurse, IDI).

## 3.3 Practices

**3.3.1 Facilitators and barriers to hand hygiene practices.** *3.3.1.1 Motivating factors for hand hygiene*. Healthcare workers in this study mentioned several facilitators to their compliance with hand hygiene standards within the hospital environment.

*3.3.1.2 Self-protection and avoiding a resurgence of Ebola*. Participants reported that they practice hand hygiene to protect themselves and their family members, patients, and the general population from healthcare-associated infections. Some nurses argued that their determination to prevent the resurgence of Ebola was one of the main reasons for which they regularly practice hand hygiene. They underlined that healthcare workers had paid a heavy price during the epidemic–that is why they believed that hand hygiene was fundamental to breaking the transmission chain of epidemic infections.

"It's because I have in mind to prevent Ebola from coming back, that's why I like washing my hands regularly, this is the main reason that motivates me". (Nursing assistant, IDI)

Healthcare workers unwaveringly placed compliance with hand hygiene standards in the context of the Ebola outbreak. For them, their good adherence to hand hygiene practices was solidly linked to the first widespread occurrence of this disease in the country.

*3.3.1.3 The existence of a hygiene committee at the hospital.* Participants pointed out that the hygiene committee played an important role in helping them comply with hand hygiene standards at the hospital. The hygiene committee organized regular training sessions and performed periodic supervision, a key factor motivating healthcare workers to be vigilant about hand hygiene.

"There is a committee here that includes several members of the hospital; this helps us a lot to practice hand hygiene". (Midwife, FGD)

*3.3.1.4 The prevention of healthcare-associated infections through hand hygiene.* Participants underlined the beneficial effects of hand hygiene on the prevention of epidemics and healthcare-associated infections. According to them, the perceived reduction of these infections in recent years was the result of healthcare workers' compliance with hand hygiene protocols in the hospitals. Specifically, all of them believed that it was due to the introduction of hand washing methods and their application in health facilities and in the communities more widely that the country had not experienced a cholera epidemic again since the Ebola epidemic.

"It's a bit an amazing remark when it comes to hand washing. If you look at the last few years before Ebola, every year we had cholera outbreaks here [in the country], but with Ebola and the introduction of hand washing methods, I think it saved us from cholera epidemics in the last few years. Some people have asked the question: 'Why, since Ebola, is there no more cholera?' And I think the answer is simple: it's hand hygiene". (Doctor, FGD)

**3.3.2 Barriers to hand hygiene practices.**   *3.3.2.1 Non-availability of hand hygiene materials.* Participants noted several barriers to hand hygiene practices. The first was the non-availability of hand hygiene materials. They argued that for a continuous hand hygiene practice, there must be a permanent supply of materials such as disinfectants, soap, towels and hand washing stations. Some of them stated that since the end of the 2013–16 Ebola outbreak, they began to notice a shortage of these materials at their facility. They attributed the shortage to the reduction and termination of the financial and material support that the country's health facilities had received from donors and partners during the Ebola epidemic.

"Often there is shortage. It's from the high level [of the hierarchy] [. . .] The way we did during Ebola, . . . I think that we must perpetuate these actions. There must not be shortage in hand hygiene materials". (Nurse, IDI)

*3.3.2.2 Lack of running water.* Another barrier was the lack of running water at the hospital. There was no water in the wards and healthcare workers had to fetch water in jerry cans and buckets from the hospital's only borehole.

"For example, at the hospital here, we have a borehole where people fetch water from. But our taps don't run. Imagine that in your office there is a tap and a reservoir [hand washing station]. But when there is no water [in the bucket], people are too lazy to go and fetch water to come and wash their hands. But if there was water from your tap, you just have to open it and wash your hands. These are things that can make it easier to sustain hand washing". (Doctor, FGD)

*3.3.2.3 Pleas for improvements.* To improve the situation at the hospital, participants solicited support from partners to ensure a supply of running water to the wards, and donations of hand hygiene materials to facilitate hand hygiene practices. They did not believe that a permanent availability of hand hygiene materials could be ensured by the hospital without support from partners. As for human resources, the participants requested continuous training, supervision and capacity building of the hospital's hygiene committee by experts. They particularly emphasized the need for training of newly employed staff. They also underlined the necessity of an operating budget to allow the hygiene committee to cope with shortages of hand hygiene materials within the hospital.

"As for the water supply, we don't yet have a partner for that; we need a partner who can fund it for us, because the resources of the hospital alone cannot solve all the problems". (Nurse, IDI)

**3.3.3 Observed practices.** During observations conducted at FRH, MD noted that hand hygiene was practiced routinely. At the hospital's main entrance, and the entrance to each ward, hand hygiene stations with 0.05% chlorinated water were installed and used for hand washing by healthcare workers, patients and anyone else coming to the facility. Posters displaying hand hygiene protocols were displayed in the wards. When attending to patients, doctors used alcohol-based hand rubs after examining the patient to disinfect their gloves before removing them, and again to clean their hands before putting on a new pair of gloves for the following patient. Some medical doctors explained good hygiene practice to their assistants, such as the necessity of changing gloves for each patient when conducting invasive procedures. Nurses/nursing assistants wore gloves and cleaned their hands before and after providing care to patients, such as giving injections.

In contrast to the abovementioned practices, MD, in two occasions, observed nursing assistants who undertook medical routines, mainly giving intramuscular injections to outpatients, without gloves and without cleaning their hands before and after care provision.

## 4. Discussion

This study examines the knowledge, attitudes and practices of hand hygiene by healthcare workers at FRH prior to the implementation of the WHO Hand Hygiene Improvement Strategy. The study identified the Ebola outbreak in 2013–16 as a pivotal element, as participants referred to this unprecedented health crisis as an event that had a major impact on their knowledge, attitudes and practices of hand hygiene. Acquired knowledge, as well as helpful attitudes and appropriate practices were attributed to various training sessions and supervision that were provided during the outbreak to stop the spread of the Ebola virus. In fact, the impact of training on infection prevention and control during an Ebola outbreak has been widely documented [26–30]. Healthcare workers became aware of Ebola's high case-fatality rate, and heard about, even witnessed colleagues dying of Ebola—this encouraged them to comply with hand hygiene standards to save their own lives and the lives of their families. This observation underlines the role that fear plays in behavior change during a health emergency [31].

Relating our findings to the thematic model deducted from a systematic qualitative review on hand hygiene by Smiddy et al. [13], it appears that healthcare workers' knowledge, attitudes and practices of hand hygiene could be determined by different factors. This model states that motivational factors such as social influences, acuity of patient care, self-protection, and the

use of cues; as well as factors relating to perception of the work environment, like resources, knowledge, information and the organizational culture, have an impact on compliance with hand hygiene practices [13].

In our study, participants reported that during the Ebola outbreak, they acquired considerable knowledge, positively changed their attitudes, and complied with good hand hygiene practices. At the height of the outbreak, the major factor that influenced their hand hygiene-related behavior was "self-protection," one of the motivational factors stated in Smiddy et al.'s thematic model [13]. Hence, their "*self-protection* conduct/behavior" could have been mainly determined by the fear of becoming infected with and dying of the Ebola virus, as was the case for some of their colleagues [32]. Indeed, participants' narratives show that the lasting effect of this fear was their determination to prevent the resurgence of Ebola. However, this seems problematic as the Ebola memory could wane, raising the concern of whether they would then go back to the old not-so-diligent-habits. As long as healthcare workers do not understand that hand hygiene allows them to avoid suffering and death in their daily practice, the message has not yet sunk in; hence the necessity for them to incorporate hand hygiene as part of their daily routines in healthcare settings.

Beyond self-protection, healthcare workers worried about others, as some of them stated that hand hygiene allows them to protect their patients, family members and the general population. Other studies have also reported that self-protection was a "consistent motivator" of hand hygiene practice, and that healthcare workers are conscious of the possibility of taking healthcare-associated infections home to their family members [33]. Healthcare workers' fear of contracting Ebola virus and possibly infecting their family members could also be related to the Health Belief Model's four components (perceived susceptibility, perceived severity, perceived benefits, and perceived barriers) [34]. Therefore, having in mind that they were susceptible to contract Ebola virus during their daily activities–and possibly spread it into their families–, perceiving Ebola virus as severe and capable to cause death (as was the case for some of their colleagues in other hospitals), and knowing the benefits of hand hygiene to prevent them (and their family members) from contracting the virus, constituted some key determinants of healthcare workers' compliance with hand hygiene practices during the outbreak.

Another motivational factor was the existence of a hygiene committee at the hospital for continuous training and supervision of hand hygiene practices. This can be counted as part of the "*social influences*" on healthcare workers as it forms a constant reminder to comply with hand hygiene practices [13]. Furthermore, a complementary motivating factor to the *"social influences"*–here the hygiene committee–could be the presence of visible resources–here photos and posters on the walls in different wards–which then constitute the *"use of cues"* [13], as they do not only serve as reminders of facts improving knowledge, but they also form attitudes.

The perceived reduction of healthcare-associated infections in general, and the non-occurrence of a cholera outbreak in the country in particular, were attributed to compliance with hand hygiene standards in health facilities after the Ebola outbreak. In fact, it has been shown that beside Ebola's negative impact, *the outbreak brought several opportunities* to the affected countries, such as the improvement of infection control skills and the capacity for disease surveillance [35], both which could also be observed in the ongoing SARS-CoV-2 pandemic [36]. However, negligence resulting in poor attitudes toward hand hygiene was noted once the EVD outbreak had ended, showing that the cue of fear–here of a deadly Ebola infection–was no longer present at the hospital and in the country. Even though healthcare workers might have knowledge on hand hygiene per their medical background and several training sessions attended during the Ebola outbreak, the absence of a frightening event such as an active infectious disease outbreak resulted in the rise of neglectful attitudes and practices toward hand

hygiene. This could also be the case in settings that have not experienced an outbreak such as Ebola, as they could fall into negligence if measures such as periodic supervisory training, monitoring and evaluation are not put in place to sustain good hand hygiene practices. These findings point to the need for sustainable approaches in health facilities to maintain good hand hygiene practices among healthcare workers, even in the absence of epidemics. In the case of the FRH, participants reported on the existence of a hygiene committee which monitors and supervises hand hygiene practices, capacity building and reinforcement. Further capacity strengthening of this hygiene committee could be envisioned to sustain hand hygiene knowledge, attitudes and practices among healthcare workers.

The study also highlights the barriers to hand hygiene practices. Major barriers at FRH were the reduction in financial and material support for hand hygiene once the Ebola outbreak had been contained, and the unavailability of running water in the wards. The lack of resources, mainly hand hygiene products such as alcohol-based hand rubs, has an impact on compliance with hand hygiene practices [36, 37]. In the case of FRH, a shortage in hand hygiene materials since the end of the Ebola outbreak hindered compliance with hand hygiene practices. Here again, the fourth component of the Health Belief Model (HBM) [34] comes into play as perceived barriers to hand hygiene practices influenced healthcare workers' attitudes towards the practice. In fact, the enumerated barriers appeared in a moment where the perceived susceptibility to contract Ebola virus waned with the end of the outbreak, resulting in neglectful attitudes towards hand hygiene practices.

Participants pleaded for financial and material support, mainly supply in hand hygiene materials and provision of running water in the wards. Indeed, some of these pleas were considered during the PASQUALE project implementation at FRH where the local production of alcohol-based hand rub was introduced to decrease the hospital's dependence on external donors and partners [10]. As some participants did not believe that the hospital could ensure the permanent availability of hand hygiene materials, sustaining this local production of alcohol-based hand rub would significantly change the mentality of heavy reliance on donors and partners.

Building on the findings of this assessment and on those of the quantitative baseline evaluation, the PASQUALE intervention was implemented [10]. A post-intervention assessment was carried out in August 2019, highlighting the impact of the project on healthcare workers' hand hygiene knowledge, attitudes and practices [11]. The use of cues was a striking factor regarding hand hygiene by healthcare workers at FRH. The local production of alcohol-based hand rubs, its availability (ready-to-use hand hygiene product) and accessibility to the hospital's personnel [10, 11], as well as several training sessions held on hand hygiene, improved knowledge, attitudes and practices among healthcare workers in general, and had a considerable impact on their hand hygiene practices within the hospital. These findings show how visible resources can enhance hand hygiene related practices among healthcare workers in a hospital setting. Indeed, evidence has shown that "cues trigger memory, attention, and decision processes and therefore trigger behavior" [36–38]. However, it has been reported that the introduction of alcohol-based hand rubs alone, with a view of decreasing hand washing efforts without any associated behavioral change program, is unlikely to induce a sustained increase in hand hygiene compliance [39].

## 5. Study limitations

This study was a qualitative baseline assessment conducted over a one-month period in order to provide an overview of healthcare workers' hand hygiene knowledge, attitudes and practices at Faranah Regional Hospital (FRH). The findings were to serve as guidelines for the

implementation of the PASQUALE project, a new partnership aimed at implementing the WHO hand hygiene improvement strategy at the hospital. This then implies, at least for the hospital's staff, some kinds of financial and material support (which was cut down at the end of Ebola outbreak) that would be brought in by the project. This context could then constitute a source of information bias. Indeed, informing healthcare workers that a researcher had arrived for a baseline assessment to start a new project could have had an influence on their behavior and answers to the questions in a way that could accelerate or facilitate the project's commencement in their health facilities. In fact, respondents' overemphasis on barriers to their compliance with hand hygiene practices could account for their desire to benefit support from external partners. Moreover, purposive sampling could also constitute a source of researcher bias as this technique highly relies on researcher's personal judgment in selecting the study participants [40]. To minimize these biases, efforts were made to triangulate data collection techniques (IDIs, FGDs and direct observations) and to ensure the representativity of all professional categories working in the hospital's different wards in the study.

## 6. Conclusion

We report qualitatively on healthcare workers' knowledge, attitudes and practices of hand hygiene at FRH prior to the introduction of the WHO Hand Hygiene Improvement Strategy. The use of cues and the existence of adequate resources appeared to be fundamental to the compliance of healthcare workers with hand hygiene practices. In the light of these findings and in order to sustainably improve hand hygiene, we recommend that hand hygiene projects focus on local capacity building through the training of healthcare workers and the provision of hand hygiene resources. More specifically, the impact of the existing hygiene committee should be strengthened by involving its members in training on hand hygiene and in the local production and management of alcohol-based hand rubs. Having a knowledgeable and active hygiene committee would contribute to a safe and sustainable hand-over after the implementation of the WHO strategy. Moreover, material support such as installing hand washing stations in the wards and ensuring running water and a reliable supply of alcohol-based hand rubs supply would enable healthcare workers to comply with WHO standards. In all cases, a high and long-term commitment of the authorities and healthcare workers is needed for a sustainable hand hygiene culture.

## Acknowledgments

We sincerely thank the late Dr Mamy Cécé Gowara, Ex-Directeur Régional de la Santé, and Dr. N'Faly Bangoura, Ex-Directeur Préfectoral de la Santé of Faranah, for their frank collaboration and facilitation of the study. We are grateful to our respondents and all health personnel at Faranah Regional Hospital for sharing their opinions with us and hosting the project in their health facilities. Finally, we thank Dr Gnèpou Ivette Mamy for her considerable help in transcribing the audio recordings made during the study, and Dr Almudena Mari Saez, Dr Isak Niehaus and Ms. Rebekah Wood for commenting on earlier versions of the manuscript.

## Author Contributions

**Conceptualization:** Moussa Douno, Sophie Alice Müller.

**Data curation:** Moussa Douno.

**Formal analysis:** Moussa Douno.

**Funding acquisition:** Matthias Borchert.

**Investigation:** Moussa Douno.

**Methodology:** Moussa Douno, Sophie Alice Müller.

**Project administration:** Ibrahima Nabe, Sophie Alice Müller.

**Resources:** Matthias Borchert, Ibrahima Nabe, Sophie Alice Müller.

**Supervision:** Matthias Borchert, Ibrahima Nabe, Sophie Alice Müller.

**Validation:** Carlos Rocha, Matthias Borchert, Ibrahima Nabe, Sophie Alice Müller.

**Visualization:** Matthias Borchert, Sophie Alice Müller.

**Writing – original draft:** Moussa Douno, Carlos Rocha.

**Writing – review & editing:** Moussa Douno, Carlos Rocha, Matthias Borchert, Sophie Alice Müller.

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
