## [Decision Letter · Decision Letter 0]

15 Nov 2022

PGPH-D-22-01633

Qualitative assessment of hand hygiene knowledge, attitudes and practices among healthcare workers prior to the implementation of the WHO Hand Hygiene Improvement Strategy at Faranah Regional Hospital, Guinea

Dear Dr. Moussa Douno,

Thank you for submitting your manuscript to PLOS Global Public Health. After careful consideration, we feel that it has merit but does not fully meet PLOS Global Public Health’s publication criteria as it currently stands. Therefore, we invite you to submit a revised version of the manuscript that addresses the points raised during the review process.

I invite you to provide MAJOR REVISIONS to your submission. We received reviews from four reviewers who provided very vital comments to improve your submission. 

We look forward to receiving your revised manuscript.

Kind regards,

Ferdinand Mukumbang, PhD

Academic Editor

Journal Requirements:

a. Please clarify all sources of funding (financial or material support) for your study. List the grants (with grant number) or organizations (with url) that supported your study, including funding received from your institution. 

b. State the initials, alongside each funding source, of each author to receive each grant.

c. State what role the funders took in the study. If the funders had no role in your study, please state: “The funders had no role in study design, data collection and analysis, decision to publish, or preparation of the manuscript.”

d. If any authors received a salary from any of your funders, please state which authors and which funders.

2. Since your data is not available for proprietary reasons, please explain via email why the data is not available. Please also include the contact information for the third party organization that should be contacted should other researchers want to request access to this data and please include the full citation of where the data can be found. We also request that you verify with us via email that any researcher will be able to obtain the data set in the same manner that the you have obtained it. If you feel you are unwilling or unable to adhere to this policy, please explain your reasons by return email and your exemption request will be escalated to the editor for approval. Your exemption request will be handled independently and will not hold up the peer review process, but will need to be resolved should your manuscript be accepted for publication. One of the Editorial team will be in touch if they require more information.

Additional Editor Comments (if provided):

Dear authors,

Thank you for submitting your manuscript to PLOS Global Public Health. Attached to this email are the comments that they provided. Please, meticulously consider these comments to revise your manuscript.

Kind regards,

Ferdinand C Mukumbang

Reviewers' comments:

Reviewer's Responses to Questions

**Comments to the Author**

1. Does this manuscript meet PLOS Global Public Health’s publication criteria? Is the manuscript technically sound, and do the data support the conclusions? The manuscript must describe methodologically and ethically rigorous research with conclusions that are appropriately drawn based on the data presented.

Reviewer #1: Yes

Reviewer #2: Yes

Reviewer #3: Yes

Reviewer #4: Partly

2. Has the statistical analysis been performed appropriately and rigorously?

Reviewer #1: Yes

Reviewer #2: Yes

Reviewer #3: I don't know

Reviewer #4: N/A

3. Have the authors made all data underlying the findings in their manuscript fully available (please refer to the Data Availability Statement at the start of the manuscript PDF file)?

Reviewer #1: No

Reviewer #2: Yes

Reviewer #3: Yes

Reviewer #4: No

4. Is the manuscript presented in an intelligible fashion and written in standard English?

Reviewer #1: Yes

Reviewer #2: Yes

Reviewer #3: Yes

Reviewer #4: Yes

5. Review Comments to the Author

Reviewer #1: This article presents a very interesting and new finding on the impact of the 2013-2016 Ebola outbreak in West Africa on hand hygiene practices, which may also provide insights for current and future pandemics. The findings are drawn from the qualitative baseline of a larger study, and as such the full data (transcripts) have not been made available. However, excerpts from the transcripts that support the author's findings are included in the results section.

Reviewer comments:

In the abstract, introduce the terms in-depth interview and focus group discussion before using the acronyms. The final sentence of the abstract speaks to the results of the assessment generally, but given that this paper is on the qualitative baseline only, it would be better to highlight only those findings. I would take the last sentence out.

Minor edits:

Abstract, page 1, line 19: remove the comma after "workers"

Abstract, page 1, line 22: take out "held"

Abstract, page 2, line 24: take out "that is"

In the introduction, please define hand hygiene best practices and explicitly link them to health care-associated infections and patient outcomes. What are some examples of the ways in which other health issues are prioritized over patient safety (line 32)? The sentence in lines 41-42 is confusing- which professional categories, duties, etc. hindered the strategy's execution, and how? The final paragraph of the introduction (lines 60-65) seems to be about the full study. It would be better to speak to the main goal of this article, which is to illuminate specific and perhaps unanticipated qualitative findings, rather than to serve as a baseline for follow-up.

Minor edits:

Introduction, page 2, line 30: take out the hyphen in "extra-costs"

Introduction, page 2, lines 37-39: Simplify this sentence by saying "The WHO strategy was first implemented on the African continent in [x year], and may have been spurred in part by the 2013-16 EVD outbreak." Specify when the strategy was implemented.

Introduction, page 3, line 58: replace "whereas research has focused rather" with "and has instead focused on" (no comma after "countries")

In the methods section, use darker colors for Table 1. In the data collection sub-section, clarify where the data collection activities took place (was it at the hospital?), as well as if anyone else was present other than the facilitator and participants (note-taker, third parties, etc). Speak to the informed consent process and how participants were recruited (face-to-face or otherwise). Was patient consent obtained for direct observations? What was the breakdown of the 13 interviews, by participant type? How many direct observations were performed? On average, data collection activities took 45 minutes. Was this average different for IDIs versus FGDs?

Minor edits:

Methods, page 4, line 90: replace "purposely" with "purposively"

Methods, page 4, line 93: replace "In whole" with "In total"

Methods, page 4, line 105: replace "entailed" with "comprised" or "included"

Methods, page 5, line 125: replace "obtained" with "received"

In the results section, lines 206-208, you mention doctors noting that lack of hand hygiene during the Ebola outbreak resulted in the deaths of healthcare workers. This finding is a stark measure of risk perception and severity. Do you have supporting quotes you could add here?

Minor edits:

Results, page 6, line 127: delete "Following the main aspects of the research,"

Results, page 7, lines 139-140: change to "ranging from washing hands with water and soap, using a 0.05% chlorine solution, to employing alcohol-based hand rubs."

Results, page 7, lines 144-145: replace "contributing to establish" with "establishing"

Results, page 7, line 152: replace "learnt a lot" with "learned more"

Results, page 7, line 153: replace "response to the crisis" with "crisis response"

Results, page 8, line 162: replace "Training on this form of hygiene" with "Hand hygiene training"

Results, page 8, line 184-185: delete "the" in the "the protocols"

Results, page 8, line 185-186: delete "the" in the "the necessary materials"

Results, page 8, line 186: add "practices" after hand hygiene

Results, page 8, line 190: delete "the passage of"

Results, page 9: line 193: You use "HH" for hand hygiene, but I don't see this acronym anywhere else. I would spell out hand hygiene if you are not going to use the acronym consistently.

Results, page 9, line 194: I would be consistent in my use of "ebola outbreak" rather than "Ebola crisis"

Results, page 9, line 195: delete "the implementation of" and "already"

Results, page 9, line 195: add "participants" after "Some"

Results, page 9, line 197: delete ").."

Results, page 9, line 200: add "towards hand hygiene" after "positive attitudes"

Results, page 9, lines 221-223: Why is this section italicized?

Results, page 9, lines 223-224: Start a new paragraph with the words in bold.

In the discussion section, when you talk about the "thematic model", please clarify which model you used by name, if it has one. You don't mention the Extended Parallel Process Model (EPPM) or the Health Belief Model but I think that both could be relevant here, and would allow you to better discuss the roles of perceived severity/perceived susceptibility as well as the "cue of fear" you have referred to previously. I think incorporating these models could make the discussion section stronger. In line 304, you mention the "use of cues" - you should connect back to this idea of the use of cues and the thematic model when you discuss the "presence of photos and posters on the walls" in the next page.

Minor edits:

Discussion, page 12, line 307: please clarify what you mean by "acuity of care"

Discussion, page 12, line 307: replace "regimes" with "routines" or "practices"

Discussion, page 12, line 318: replace "every single normal day" with "in their daily practice"

Discussion, page 13, line 345: replace "This points" with "These findings point"

Discussion, page 13, lines 347-348: replace "is committed to monitor and supervise" with "monitors and supervises"

Discussion, page 13, line 354: replace "have an impact" with "has an impact"

Discussion, page 14, line 355: take out "HH", consider another word other than "regimes"

Discussion, page 14, line 356: replace "was seen as a hindrance to the compliance" with "hindered compliance"

Discussion, page 14, line 356: delete "in this view"

Discussion, page 14, line 357: specify which materials participants were asking for

Discussion, page 14, line 365: delete "Afterwards" (this is clear from "post-intervention"

Discussion, page 14, line 372: replace "This shows" with "These findings show"

Reviewer #2: PLOS GPH article review:

Manuscript Number PGPH-D-22-01633

"Qualitative assessment of hand hygiene knowledge, attitudes and practices among healthcare workers prior to the implementation of the WHO Hand Hygiene Improvement Strategy at Faranah Regional Hospital, Guinea"

Authors:

Reviewed by: Suresh Taman, PhD

Word consistency:

Health care vs. healthcare

Methodology:

Looks like it is a longitudinal follow up study. If so, mention it.

Mention no of observation, IDI and FGDs in the methodology section

If it is a purposefully sampled respondents, specify the inclusion and exclusion criteria.

Reviewer #3: Lines 41-42: Unclear relationship of how certain professional categories, duties, hospital areas, and services hindered the strategy’s execution. Was it that those certain categories, duties, areas, and services were associated with lower uptake of the strategy? Any trends relevant in those association that apply to FRH?

Line 74: Have you triangulated and reported both the quantitative and qualitative data together in addition to the quantitative data separately? Demonstrating how the quantitative and qualitative data corroborate or contradict each other would strengthen the findings.

Line 90: possible typo: should purposely be purposively?

Lines 91-97: How did you purposively select interviewees and FGD participants? Please also explain why sampling was done purposively as opposed to randomly. How was it determined whether respondents would participate in an FGD as opposed to an IDI? What was the method used for recording the observations?

Methods overall: Please include a limitations section. In this section, please also address whether participants knew they were being observed and any influence that may have had on behavior.

Lines 138 – 150: Consider separating out finding on knowledge of methods and practices vs knowledge that hand hygiene contribute to quality care

Lines 221 – 223: change in formatting with italic text was distracting

Lines 274 – 287: How was the observation data recorded? Is there additional or more specific data on the number of incidents of HH? It is mentioned that doctors practiced HH while some nurses/nursing assistants did and some did not – is there a more specific breakdown of HH practices observed by type of employee?

Results overall: I think the results would be strengthened by including the quantitative data from the baseline to better understand how health workers knowledge, attitudes, and beliefs reinforces or contradicts reported and observed behaviors.

Lines 301 – 307: Did this theoretical model inform the development of study design and data collection tools? If so, specify in the methodology section

Line 365: Describe in more detail in what ways PASQUALE responded directly to the findings of this study and the quantitative baseline

Overall: please include copies of the interview and FGD tools as an annex

Reviewer #4: This paper reports on healthcare providers’ hand hygiene knowledge, attitudes, and practices prior to implementation of the WHO strategy at Faranah Regional Hospital. The qualitative exploration of knowledge, attitudes, and practices holds value for informing efforts to prevent healthcare-associated infections. However, the manuscript includes limited methodological detail, making it challenging to determine the methodological rigor of the study. Additionally, the findings are not clearly delineated and would benefit from significant reorganization.

Below are specific recommendations that could help strengthen the paper:

1. More detail on the observation procedures, including how many people were observed, with what frequency, and the total duration of the observation, would be useful. Additionally, how were the observation data documented and analyzed?

2. The authors reference thematic analysis, however, there is no description of the analytic process beyond the development of a codebook. Please detail how transcripts were coded (e.g. in duplicate?, how were discrepancies resolved?), how coded data were then analyzed and organized into themes, and any rigor steps taken to strengthen the analysis.

3. The authors mention data saturation, but do not describe how they assessed saturation or whether the sample size was driven by, or independent of, saturation. Please describe how and when saturation was addressed, e.g. were data collection and analysis conducted concurrently or was saturation assessed after the fact?, and what approaches as used to determine the point at which saturation was achieved?

4. The authors state that the results are presented in three categories: knowledge of hand hygiene, attitudes toward hand hygiene, and hand hygiene practices. However, the results are presented in four sections: (1) knowledge, (2) attitudes, and (3) barriers and facilitators to hand hygiene, which also includes suggested improvements; and (4) hand hygiene practices. Clearly outlining the presentation of findings will help orient the reader.

5. The knowledge, attitudes, and barriers and facilitators sections lack clear differentiation. For example, the findings related to ‘learning from the ebola epidemic,’ and ‘the influence of the ebola epidemic,’ are not meaningfully differentiated, and the section on the influence of the ebola epidemic, in particular, weaves between how healthcare workers became knowledgeable about hand hygiene, and how their attitudes toward hand hygiene were affected by the 2013-2016 Ebola outbreak. Additionally, the section on facilitator and barriers to hand hygiene practices weaves between priorities and values that shape the attitudes of healthcare works (e.g. self protection, avoiding resurgence) and tangible things like the presence of a hospital hygiene committee or the absence of soap, running water, etc. The manuscript would benefit from a restructuring of the presentation of findings to address areas of overlap and improve clarity of the major themes and sub-themes.

6. Did the authors consider differences in knowledge and attitudes based on profession? It would be interesting to explore and address any differences between medical doctors, midwives, nurses, and nursing and laboratory assistants. Additionally, as written, the illustrative quotations disproportionally represent doctors’ views, while midwives, nursing and laboratory assistants’ voices are not well represented. The manuscript would be strengthened by the addition of more representative quotes, either in the text or in a supplementary table.

7. The observed practices section would benefit from some quantification. For example, how frequently was hand washing practiced versus not practiced by nurses and nursing assistants? As currently written, this section is not enhancing the manuscript.

6. PLOS authors have the option to publish the peer review history of their article (what does this mean?). If published, this will include your full peer review and any attached files.

**Do you want your identity to be public for this peer review?** For information about this choice, including consent withdrawal, please see our Privacy Policy.

Reviewer #1: No

Reviewer #2: **Yes: **Surehs Tamang, PhD

Reviewer #3: No

Reviewer #4: No

---

## [Editor Report · Decision Letter 1]

17 Jan 2023

Qualitative assessment of hand hygiene knowledge, attitudes and practices among healthcare workers prior to the implementation of the WHO Hand Hygiene Improvement Strategy at Faranah Regional Hospital, Guinea

PGPH-D-22-01633R1

Dear Dr Douno,

We are pleased to inform you that your manuscript 'Qualitative assessment of hand hygiene knowledge, attitudes and practices among healthcare workers prior to the implementation of the WHO Hand Hygiene Improvement Strategy at Faranah Regional Hospital, Guinea' has been provisionally accepted for publication in PLOS Global Public Health.

Best regards,

Ferdinand Mukumbang, PhD

Academic Editor